# On the price dynamics of non-fungible tokens: The 'Bored Apes' case

**Roberto Cellini** [iD]*, **Tiziana Cuccia, Johan Lyrvall**

Department of Economics and Business, University of Catania, Catania, Italy

* cellini@unict.it

## Abstract

We analyse the pattern of daily price of a collection of artistic non-fungible tokens, namely, the "Bored Ape Yacht Club" (BAYC) collectibles, over the first year of their life, from May 2021 to May 2022. Taking a time-series analysis approach, we consider the daily average price, and other variants of daily price index, derived from hedonic regression model. Aesthetic features of the collectibles do matter. At the same time, the price series emerge to be non-stationary, integrated of order 1, with their first difference exhibiting heteroscedasticity and autoregressive variance. Models of ARCH/GARCH class are appropriate to describe the dynamics. Though the price series of BAYC collectibles and their daily movements share many characteristics with the series of financial assets, they do not appear to be related to financial variables from both the crypto- and the real (i.e., not crypto) world.

## 1. Introduction

This study aims to investigate the features of the price of the pieces of the *Bored Ape Yacht Club* (BAYC) collection, which is one of the most known Non-Fungible Token (NFT) collection with artistic content.

A NFT is a unit of code on a blockchain that refers to a unique digital or physical asset like a virtual artwork, an audio-visual recording, or a real-estate transaction. Any given NFT is stored on a blockchain, and it provides its owner with inherent proof-of-ownership and proof-of-originality. As such, NFTs are scarce, irreplaceable, and un-interchangeable assets.

Most culturally significant NFT collections–including the BAYC collection–belong to a category commonly referred to as "profile picture projects" due to the common practice of owners to put their NFT as their profile picture on Twitter. These projects consist of a fixed set of recognisable, aesthetically uniform and unique avatars. In the specific case of BAYC, the collection consists of 10,000 digital collectibles of "Bored Apes", each of which characterised by a unique combination of traits (clothes, headwear, colours, expressions, etc.) from a given menu. Traits and their rarity (scarcity) are inherent features of the collectibles. Bored Apes serve as an owner's digital identity, and the Bored Ape owners can gain other "exclusive benefits" from their NFT–substantially, permitting the participation in reserved online meetings and chat.

Though NFTs represent a young market, an economic literature body is still existing, with special attention to the determinants of price and volume of transactions. Our present study

**Data Availability Statement:** All data (raw data) are freely downloadable from https://opensea.io and from https://www.nft-stats.com/collection/boredapeyachtclub. Additionally, data relevant to this study are available in the Supporting Information files.

**Funding:** The author(s) received no specific funding for this work.

**Competing interests:** The authors have declared that no competing interests exist.

aims to contribute to this growing body of literature, providing an analysis on the price pattern of one of the most traded collection in the NFT world.

The general question we aim to answer is whether collections like BAYC are closer to financial assets or artistic items. More specifically, we aim to answer the following research questions: (1) Does the time pattern of the prices of "artistic" BAYC NFT collectibles share characteristics with the time pattern of "traditional" financial assets (e.g., equities or exchange rates)? (2) What are the characteristics of the daily returns of these collectibles?–Do they contain autoregressive heteroscedasticity? Is there daily seasonality? (3) Is there any relation between the daily prices of these collectibles and financial markets?; and finally, (4) Do the aesthetic characteristics of the collectibles matter for the price determination?

Answering these questions is important not only *per se*, in order to collect evidence on a specific case, but also to shed light on the nature of the NFTs. To better understand the contribution of this paper to the literature, it is appropriate to provide a background of the specific case study, and a short review of the relevant economic literature.

## Background: The case study

The NFT collection *Bored Ape Yacht Club* (BAYC), minted by a group of four anonymous people, was launched in April 2021, with sales starting on May 1st, 2021. It is hosted on the Ethereum blockchain and transacted mainly on the NFT platform OpenSea, with 24/7 accessibility. After the launch, the collection quickly became a pop-cultural institution, attracting influential icons such as the basketball player Stephen Curry, the rapper Eminem, not to mention Elon Musk, who changed his Twitter profile picture to a collage of the BAYC NFT images for some days in May 2022. In May 2022, one year after the launch, the members of the club were about 6,300 (the last available datum, April 2023, is around 6,000). Payments are made primarily through Ether (ETH), one of the most used cryptocurrencies. Of course, the price of BAYC NFTs can be also expressed in US Dollar (USD, $) or Euro (EUR), using the appropriate exchange rate.

According to press and websites, the four anonymous founders of BAYC spent around $40,000 to create and to launch the collection. Prices of BAYC collectibles have been rocketing over the first year of life–especially over the first semester. Just for the sake of providing some data, consider that the collectibles have increased floor price from ETH 0.1 in May 2021 to ETH 120 in April 2022, that is, from about EUR 870 to over EUR 300,000. The average price of sold BAYC NFTs moved from ETH 0.64 in May 2021 to ETH 129.14 in April 2022, i.e., from EUR 1,650 1 to EUR 368,350. The highest paid price for a single BAYC NFT piece, on the OpenSea platform, is ETH 1,080, corresponding to USD 2.8 million in January 2022; however, the record price for a single piece, 3.4 million USD, was reached in a special auction held by the digital Sotheby's Metaverse, outside the OpenSea platform, in October 2021.

Weekly trading volume–with peak around USD 65 million (corresponding to 265 sold collectibles, with average price of $247,000) in January 2022– has dropped significantly in subsequent months, mainly due to a decrease in the number of sales, rather than a change in average price, until May 2022. In the second semester of 2022, outside the time sample under consideration in the present investigation, also prices have decreased and ETH devaluated against USD and EUR (in December 2022, weekly trading volume was around USD 24 million, corresponding to 237 sold collectibles with average price around $101,000; last available information, referred to April 2023, provide weekly trade volume around USD 26 million, corresponding to 250 sold collectibles with average price around $104,000; data from https://www.nft-stats.com/collection/boredapeyachtclub).

The aggregate value of the collection and its increase and variability over the first year of life make the BAYC collection an interesting case to study.

## The literature: A short review

NFTs represent a young market, and the scientific attention on their economic aspects is obviously recent. Available contributions study specific cases or segments of the NFT markets, paying attention to the determinants of price and their patterns, and to the links with other markets, in the crypto- and in the real world. Borri, Liu and Tsyvinski [1] and Kraeussl and Tugnetti [2] provide excellent and comprehensive literature reviews. Among the recent analyses, we mention the following contributions, relevant to the present paper.

Dowling [3] studies the pricing of three NFT markets (*CryptoPunks*, *Decentraland*, and *Axie Infinity*) and the relationship with cryptocurrency markets. Evidence suggests that there is only limited volatility transmission between cryptocurrency and NFT markets, and also limited correlation between different items in the NFT world. Kong and Lin [4] focus on one of earliest collections, the *CryptoPunks*, obtaining a monthly price index, for the period 2017–22, based on hedonic regression model; they find that the aesthetic characteristics of the images (like the presence of helmet, cap, beard) do affect prices. Differently from [3], Kong and Lin [4] find that their index positively co-moves with the exchange rate of the cryptocurrency and the US Dollar. In their analysis on NFT *CryptoPunk* prices, Kong and Lin [4] arrive at the conclusion that these items are highly speculative, but investors are ready to spend money on them, due to the "emotional dividends" given by their possession. Nguyen [5], extending [4], and focusing on the role of aesthetic characteristic of the pieces upon their price, shows that *CryptoPunks* with lighter skin tones have higher prices than the ones with dark skin. At the same time, prices are positively correlated with the exchange rate of cryptocurrencies with US dollar and negatively correlated with the exchange rate volatility. In general, returns of investment in *CryptoPunks* are higher and more volatile as compared to other asset classes (including a number of stock and bond indices) and commodities (such as gold, real estates, paintings and wines). Nadini et al. [6] study the extent to which different categories of NFTs (e.g., collectibles, artworks, virtual game assets, virtual properties in the Metaverse, etc.) contribute to the whole NFT market size, and study the distribution of NFT prices across categories. They investigate the predictability of NFT sales and show that sale history provides good predictors for price dynamics. Interestingly, they also show that traders typically specialize on NFTs with similar objects and similar visual features, and form closed cluster. Kireyev and Lin [7] present a structural model with an auction game structure and show that buyers value NFTs (specifically, *CryptoKitties* in 2019) in line with expected price, while sellers have a tendency to price sub-optimally.

All these studies aim to uncover how prices of NFTs are determined. Through different methodological approaches, they suggest that NFTs have low correlation with both the existing asset classes–e.g., equity, commodity and currency–and cryptocurrency. In general, a positive correlation between NFT prices and the exchange rate of cryptocurrency emerges only over sufficiently long period of time, i.e., time sample covering the simultaneous growth of all parts of the crypto world.

## The contribution of the present study

Our present analysis expands the studies on NFTs, focussing on daily prices of BAYC NFT collectibles, which are among the most traded crypto-activities. It sheds light on the nature of non-fungible tokens with aesthetic contents. In particular, it contributes to the literature in four ways.

First, our study highlights specific characteristics of the daily price index of BAYC NFTs, within a time series analysis framework. Specifically, we show that the daily prices of BAYC NFTs–like the prices of many financial assets–are non-stationary (i.e., the time series is integrated of order 1 and shocks have permanent effect) and the daily returns are highly volatile and heteroscedastic with autoregressive variance. Thus, the daily return series may be appropriately modelled by ARCH/GARCH models, largely used in analysis of financial markets.

Second, we aim to assess whether day effects in daily return–which, to the best of our knowledge, are not yet studied by available literature on NFT–do play a role in both the mean and the variance part of the model. Our investigation shows that the day effects play a limited role, in both parts of the model; this evidence is common to financial assets in recent years.

Third, we investigate the correlation of BAYC NFT returns with other assets in both the crypto- and the traditional (i.e., not crypto-) markets; available literature, concerning different case-studies, provides mixed evidence in this respect. In our present analysis, correlation emerges to be low.

Finally, we show that aesthetic features of the collectibles do matter for the price determination. Specifically, we consider the prices conditional on the characteristics of the exchanged pieces in a hedonic-price framework. We find that that the traits of the images–like hair colour, eyes, clothes, which are elements of differentiation that can be also appreciated from an artistic/aesthetic point of view–do matter.

Thus, these items of BAYC are both collectibles and unique compositions that represent a new medium for artistic expression within the online "crypto art" field. At the same time, they are investment assets and the price dynamics share relevant features with the price dynamics of financial assets. Let us notice that the multidimensional nature of BAYC NFTs makes it difficult to define an "intrinsic value": on the one side, they provide economic returns on which an intrinsic financial value could be measured; on the other side–as noted also by Kong and Lin [4]–the owners derive an emotional dividend from the possession during the holding period, since they can use these assets to signal their social status and obtain social recognition; moreover, like all arts items, BAYC images can have an existence value *per se*, independent of individual preferences, specific usage, and financial returns.

The paper is structured as follows. Section 2 presents the theoretical framework and the research design. Section 3 presents the data and their basic statistical properties. Section 4 presents the main results–specifically, the results from the univariate analysis based on ARCH/GARCH models; the analysis of the daily seasonality in the level and volatility of the series; the relation of BAYC NFT price with financial variables, from the crypto and the real world. Section 5 considers an alternative price index derived from a hedonic price approach; the obtained evidence can be interpreted as a robustness check of the main analysis, but it also provides elements for evaluating whether the aesthetic characteristics of the pieces of BAYC collection do matter in affecting prices. Section 5 concludes.

## 2. Theoretical framework and the research design

We aim to establish whether the dynamics of the prices of BAYC NFTs share features with the typical dynamics of financial assets. We substantially follow a univariate model analysis approach, focussing on the daily series of BAYC prices.

Common features of daily data of financial series–such as the prices of assets and equities, bilateral exchange rates, and also gold, energy, and oil products–are non-stationarity (specifically, integration of order 1) and heteroscedasticity. Studies typically deal with variables in log, and find that the first-difference series of such financial variables are stationary, have little autocorrelation, but present a strong heteroscedasticity: see the recent analyses of Ahmed and

Huo [8], and the reference therein; see also the comprehensive, though not up-dated, review by Bollerslev [9] and Sarno and Taylor [10, especially Chapters 2 and 3] and Baillie and Bollerslev [11, 12], whose seminal approach we specifically follow here, and Hsie [13]. These pieces of evidence mean that shocks on price have permanent effects and the variability of price movements is autoregressive.

In these cases, models with a time dependent conditional heteroscedasticity–that is, models of class ARCH (AutoRegressive Conditional Heteroscedasticity)–are appropriate to describe the time dynamics of prices (Sarno and Taylor, [10]; Cheung et al. [14], Borreslev [15]).

Denoted by *LP* the log of the price of an asset, and by *DLP* its first difference, the basic ARCH model is:

$$\begin{cases} DLP_t = \alpha_0 + \varepsilon_t; \ \ \varepsilon_t \approx \Gamma(0, h_t) \\ h_t = \omega_0 + \sum_{i=1}^{q} \gamma_i \varepsilon_{t-i}^2 + \eta_t \end{cases} \tag{1}$$

where $\varepsilon_t$ denotes a stochastic process which follows a (non-normal) $\Gamma$ distribution with nil mean and moving variance $h_t$; $\eta_t$ denotes a white-noise error term; $q$ denotes the order of the autoregressive process of the variance of the error term. The first equation of system (1) represent the "mean part" of the model, while the second equation represents the "variance part" of the model. If $q = 1$, the variance of the error terms follows a AR(1) process. If the order of autocorrelation of the variance is higher, the model can be rewritten, in a more general and parameter-parsimonious way, in the so-called GARCH($q,p$) version:

$$\begin{cases} DLP_t = \alpha_0 + \varepsilon_t; \ \ \varepsilon_t \approx \Gamma(0, h_t); \\ h_t = p\omega_0 + \sum_{i=1}^{q} \gamma_i \varepsilon_{t-i}^2 + \sum_{j=1}^{p} \lambda_j h_{j-i} + \eta_t \end{cases} \tag{2}$$

Daily data measuring the first difference of the log price can be easily interpreted as the daily returns.

The model can be improved, exiting from a strictly univariate analysis approach, by inserting further exogenous explanatory variables. For instance, daily time effects can be inserted. Indeed, financial series, notably asset prices and exchange rates, may show daily seasonality, that is, statistically significant day effects in the mean and/or in the variance part of the model. Day effects in the first equation of the models mean that the values are, on average, lower or higher in specific days of the week; day effects in the variance part of the model mean that the variability is larger or smaller in specific days. As a matter of fact, the statistical significance of day effects in financial series was pretty common in data from the last decades of the 20[th] century; they are less common in investigations on more recent data. Increased volatility around weekends and vacation days in financial markets was found by several studies in the 1980s, see, e.g., Keim and Strambaugh [16] and French and Roll [17]. In Baille and Bollerslev [11, p. 63], the evidence of higher variance at Monday, regarding a set of bilateral exchange rates, is related with the fact that information flow after two vacation days is particularly relevant, and this leads to higher variability. However, the relevance of day effects is questioned by many authors, for cases pertaining to more recent time periods: see, e.g., Yamori and Mourdoukow [18], Yamori and Kurihara [19], Cho et al. [20] and Cellini and Cuccia [21], especially as far as the possible 'disappearance' of vacation day effect or weekend effects in financial series over the last years.

Daily seasonality in NFT price has not yet investigated by available literature, to the best of our knowledge, and the possible presence of such a component is worth investigating.

After the univariate study of the characteristics of the daily time series of the BAYC NFT prices, we move to evaluate possible relations with financial markets, and specifically with the

pattern of equity indexes, exchange rates and commodity prices. We have already reported that available evidence related to different NFTs provide mixed results concerning the relation with financial markets. Since all series under investigation are non-stationary and integrated of order 1, the cointegration tests provide a natural language to evaluate the presence of long- and/or short- run links. We will also evaluate the presence of Granger-causality links. The results of the present analysis–which resorts to well established statistical tools, not yet used by available analyses of NFT prices–lead to a clear-cut conclusion, supporting the lack of robust links between the dynamics of BAYC NFT prices and traditional financial markets.

Schematically, in our research design, we firstly check whether the daily series of BAYC price are non-stationary, with auto-regressive and heteroscedastic first-difference. Second, we check whether ARCH/GARCH models are appropriate to describe its dynamics. Third, we investigate the possible presence of daily seasonal effects in the mean and the variance part of the model. Fourth, we evaluate whether BAYC NFT prices are related with financial market dynamics. Finally, we build a price index for BAYC collection, taking a hedonic price approach; this exercise is interesting per se, as it provides evidence concerning the effects of the aesthetic features of the collectibles upon their price, and can be also interpreted as a robustness check on the results obtained in the main analysis.

## 3. Data

Our databank is computed, starting by the price and quantity of BAYC NFTs (available at https://opensea.io/). We firstly consider the simple average price of traded collectibles in any day, in ETH; this variable is denoted by $P_{ETH}$. The plot of the $P_{ETH}$ series is in Fig 1, while Table 1 reports some descriptive statistics concerning $P_{ETH}$ (with its log, $LP_{ETH}$, and the first difference $DLP_{ETH}$), the price in Euro ($P_{EUR}$; using the daily closing exchange rate) and the number of daily sales ($N_{SALES}$). Sales started on Saturday May 1 st, 2021, but since unusual values of price and quantity are registered in the first two days of sales, our databank covers the

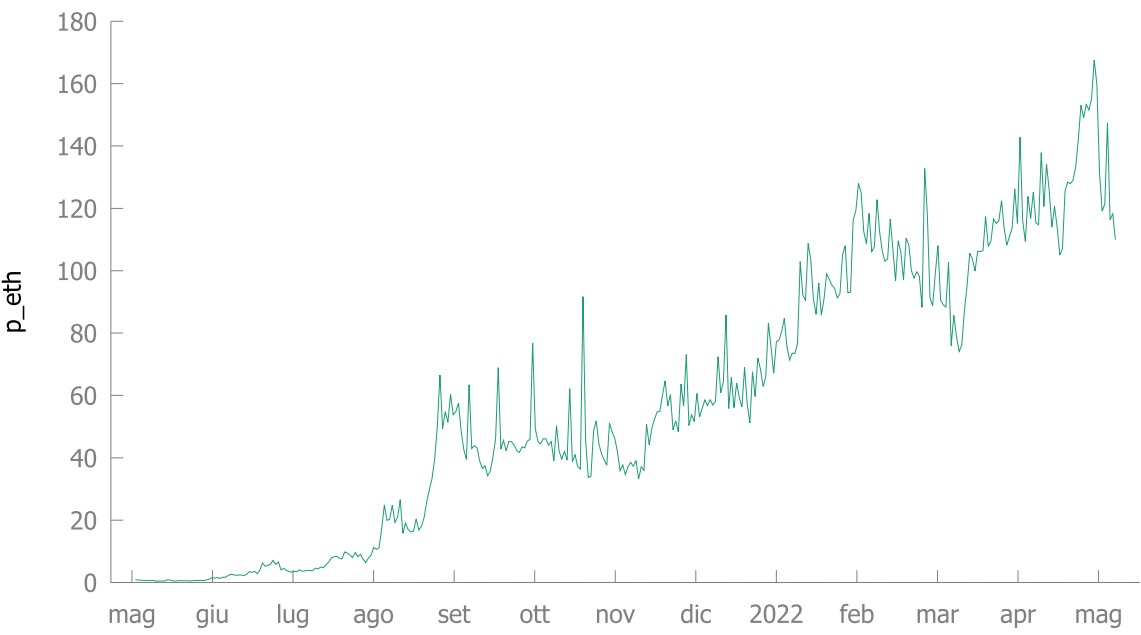

**Fig 1. BAYC-NFT prices (May 3, 2021-May 8, 2022).**

**Table 1. Descriptive statistics for average daily price and quantity.**

| | $N_{SALES}$ | $P_{ETH}$ | $LP_{ETH}$ | $DLP_{ETH}$ | $P_{EUR}$ |
|---|---|---|---|---|---|
| **Sample: May 3, 2021 –May 8, 2022 (obs 371; 370 for $DLP_{ETH}$)** | | | | | |
| Mean | 60.180 | 57.556 | 3.368 | 0.0132 | 165,791.6 |
| Median | 25.000 | 51.121 | 3.934 | 0.007 | 177,603.2 |
| Maximum | 1120.0 | 167.624 | 5.122 | 0.924 | 445,512.9 |
| Minimum | 2.0 | 0.394 | -0.931 | -0.701 | 849.6 |
| Std. Dev. | 102.112 | 43.729 | 1.596 | 0.171 | 121,292.8 |
| Skewness | 4.888 | 0.3088 | -1.238 | 0.232 | 0.110548 |
| Kurtosis | 38.711 | 1.973 | 3.392 | 6.514 | 1.953353 |

Note: See text for variables' description. $DLP_{ETH}$ includes 370 observations. All computations are provided by EViews econometric package

period from Monday May 3$^{rd}$, 2021 to Sunday May 8$^{th}$, 2022; it is made by 371 daily observations, corresponding to 53 complete weeks.

It is clear, also from a quick look at the plot in Fig 1, that the price series is non-stationary and heteroscedastic; formal statistical tests confirm these conclusions.

i. As far as stationarity and integration of the series concern, the ADF and PP statistics (in Table 2) lead to the conclusion that the price series is I(1), (i.e., integrated of order 1): $P_{ETH}$ is non-stationary, while its first difference is stationary. Integration of order 1 means that daily shocks on BAYC-NFT price have permanent, rather than temporary, effect. This results also holds in specific sub-periods (e.g., in any of the sub-period before and after August 31, or before and after October 31, 2021). Also the series in *log* is I(1), and the same results hold for the series of price expressed in Euro or in US Dollar.

ii. As far as the heteroscedasticity of the BAYC NFT price series concerns, Table 3 reports the first 10 autocorrelation coefficients, along with the Box-Pierce Q statistics for $DLP_{ETH}$ variable and its squares: the squared data exhibit substantially more autocorrelation, suggesting the presence of conditional heteroscedasticity.

The standard test procedure to evaluate the presence of autoregressive heteroscedasticity leads to the conclusion that heteroscedasticity is autoregressive indeed: We consider the residuals from the Ordinary Least Square (OLS) regression of the first difference of the price index (in log) of the BAYC NFTs ($DLP_{ETH}$) on the constant term, and test for a autoregressive process concerning the squared residuals. Test statistics for AR(1) process gives $F_{1,367} = 45.767$

**Table 2. Test for series stationarity.**

| Variable | ADF | PP | Variable | ADF | PP |
|---|---|---|---|---|---|
| $P_{ETH}$ | $t = -0.945$ ($p = 0.773$) | $t = -1.396$ ($p = 0.585$) | $DP_{ETH}$ | $t = -14.074$ ($p = 0.000$)*** | $t = -29.679$ ($p = 0.000$)*** |
| $LP_{ETH}$ | $t = -2.421$ ($p = 0.137$) | $t = -2.187$ ($p = 0.211$) | $DL\,P_{ETH}$ | $t = -25.760$ ($p = 0.000$)** | $t = -26.487$ ($p = 0.000$)*** |
| $P_{EUR}$ | $t = -1.319$ ($p = 0.622$) | $t = -1.789$ ($p = 0.386$) | $DP_{EUR}$ | $t = -15.843$ ($p = 0.000$)** | $t = -31.588$ ($p = 0.000$)*** |
| $LP_{EUR}$ | $t = -2.160$ ($p = 0.222$) | $t = -1.955$ ($p = 0.307$) | $DL\,P_{EUR}$ | $t = -17.285$ ($p = 0.000$)** | $t = -26.569$ ($p = 0.000$)*** |

Note: ADF and PP indicate, respectively, the augmented Dickey-Fuller and the Phillips-Perron test in which the null is the presence of a unit root in time series. Model selection, where appropriate and not differently specified, is based on the Schwarz information criterion. In PP test procedure, Bartlett kernel is used to select the spectral estimation method and Newey-West procedure is used to select the bandwith. *,**,*** denote significant at the 10%, 5%, 1% level, respectively. Sample from May 3, 2021 to May 8, 2022 (371 obs.), adjusted according to considered lags.

**Table 3. Autocorrelation of DLP$_{ETH}$—raw and squared data.**

| | $DLP_{ETH}$ | | | | $DLP_{ETH}$_Squared | | | |
|---|---|---|---|---|---|---|---|---|
| | **AC** | **PAC** | **Q-Stat** | **Prob** | **AC** | **PAC** | **Q-Stat** | **Prob** |
| 1 | -0.288 | -0.288 | 30.920 | 0.000 | 0.326 | 0.326 | 39.528 | 0.000 |
| 2 | -0.029 | -0.122 | 31.230 | 0.000 | 0.016 | -0.101 | 39.619 | 0.000 |
| 3 | -0.048 | -0.102 | 32.081 | 0.000 | -0.012 | 0.017 | 39.675 | 0.000 |
| 4 | 0.028 | -0.024 | 32.381 | 0.000 | 0.144 | 0.163 | 47.469 | 0.000 |
| 5 | 0.087 | 0.088 | 35.218 | 0.000 | 0.102 | -0.004 | 51.361 | 0.000 |
| 6 | 0.038 | 0.104 | 35.761 | 0.000 | 0.043 | 0.021 | 52.073 | 0.000 |
| 7 | -0.005 | 0.066 | 35.771 | 0.000 | -0.015 | -0.023 | 52.159 | 0.000 |
| 8 | -0.109 | -0.079 | 40.259 | 0.000 | -0.009 | -0.014 | 52.187 | 0.000 |
| 9 | 0.015 | -0.049 | 40.350 | 0.000 | -0.012 | -0.021 | 52.241 | 0.000 |
| 10 | -0.013 | -0.063 | 40.415 | 0.000 | 0.042 | 0.048 | 52.902 | 0.000 |
| 35 | -0.009 | -0.035 | 62.720 | 0.003 | -0.020 | -0.011 | 74.807 | 0.000 |

Note: Sample from May 3, 2021 to May 8, 2022 (371 obs), adjusted according to considered lags.

($p$ = 0.000), $LM(1)$ = 40.914 ($p$ = 0.000). The conclusion that autocorrelation is of order 1 is supported by appropriate test on higher order autocorrelation, which cannot reject that higher-order autocorrelation coefficients are not statistically significant (we checked up to 7 lags; some doubts can arise for the 4th-order lag, but all information criteria drive to select the model with 1 lag). Autoregressive heteroscedasticy means, from an economic point of view, that the variance of the returns is not constant over time, and volatility induces subsequent volatility.

## 4. Empirical findings

### A model for the BAYC NFT price dynamics

Provided that the BAYC NFT price series (and its log) has a unit root, we model the dynamics of its first difference. As already mentioned, the first difference of the daily prices in log can be interpreted as the daily returns. Since the variance of the first difference series is not constant, we consider a model with a time dependent conditional heteroscedasticity, such as the ARCH model (1), or the generalised version (2) which, in the case at hand, are rewritten as:

$$DLP_{ETHt} = \alpha_0 + \varepsilon_t; \quad \varepsilon_t \approx \Gamma(0, h_t); \quad h_t = \omega_0 + \sum_{i=1}^{q} \gamma_i \varepsilon_{t-i}^2 + \eta_t \tag{1}$$

$$DLP_{ETHt} = \alpha_0 + \varepsilon_t; \quad \varepsilon_t \approx \Gamma(0, h_t); \quad h_t = \omega_0 + \sum_{i=1}^{q} \gamma_i \varepsilon_{t-i}^2 + \sum_{j=1}^{p} \lambda_j h_{j-i} + \eta_t \tag{2}$$

In the empirical analysis of the BAYC NFT price (in first-difference of log), we adopt a "from general to particular" model selection strategy: we start from the evaluation of a ARCH(7) model, and subsequently proceed to drop statistically insignificant terms; the appropriate specification turns out to be ARCH(4). In the evaluation of the GARCH specification, starting from the general case of $p$ = 7, $q$ = 7, and subsequently dropping insignificant terms, we end up with the appropriate specification GARCH (1,1)–see Table 4. Information criteria are very similar (and not unanimous) between the ARCH(4) and GARCH(1,1) models, and we prefer to look at the GARCH(1,1) specification as the baseline model, as usual for many financial daily time series. In this paper, we present only the results from "standard" GARCH models,

**Table 4. ARCH/GARCH regression model.**

| Dependent variable: $DLP_{ETH}$ | | |
|---|---|---|
| Sample: May 4, 2021- May 8, 2022 | | |
| **MODEL:** | **ARCH(4)** | **GARCH(1,1)** |
| Mean equation | | |
| C | 0.013 (1.73)* | 0.018 (2.29)** |
| Variance equation | | |
| C | 0.031 (13.50)*** | 0.003 (2.89)*** |
| RESID(-1)_squared | 0.241 (3.16)*** | 0.176 (3.92)*** |
| RESID(-2)_squared | -0.022 (-0.73) | |
| RESID(-3)_squared | -0.064 (-3.21)*** | |
| RESID(-4)_squared | 0.103 (5.28)*** | |
| GARCH(-1) | | 0.721 (11.26)*** |
| Diagnostic | | |
| S.E. of regression | 0.172 | 0.172 |
| Log likelihood | 158.68 | 148.97 |
| Durbin-Watson stat | 2.575 | 2.572 |
| Akaike info criterion | -0.825 | -0.784 |
| Schwarz criterion | -0.762 | -0.741 |

Note: $z$-stat in parenthesis; Starred coefficient are significant at the 1% (***), 5% (**) or 10% (*) level. Computations are made through EViews software package.

and we do not deal with the problem of which variant of GARCH model could be the best. In general, the answer depends on the specific purpose of the analysis. In the case at hand, variants like EGARCH or PARCH do not provide substantially different evidence, and there is no univocal compelling answer to the question about the best variant. (Caporale and Zekokh [22] and Grobys [23] are examples of empirical investigations, concerning cryptocurrencies, where a comparative evaluation of different variants of GARCH models is presented).

In the model specifications under present consideration, the size of coefficients of the autoregressive terms–in particular, $\sum_i \gamma_i < 1$, in the ARCH(4) model, and $\gamma_1 + \lambda_1 < 1$ in the GARCH (1,1) model–supports the fact that variance is autoregressive and stationary, that is, the volatility of returns has some persistency, but it is not explosive.

So, the conclusion is that the price of BAYC NFT follows a statistical process which is very common for variables in financial markets, equity prices, equity indexes, and exchange rates (Campbell and MacKinlay [24]; Brooks [25]).

## Daily seasonality in BAYC NFT price level and volatility

As already mentioned, daily seasonality in NFT price has not yet investigated by available literature, even if the analysis of daily seasonality is a common exercise for financial assets. For a preliminary statistics evaluation, we group the data by days and test whether mean, variance and median values are different across the days of the week. Results are in Table 5. No significant statistical difference across days emerges, from a descriptive statistics point of view.

The conclusion is largely confirmed, also by regression models. We evaluate the significance of weekday fixed effects in ARCH(7), ARCH(4), ARCH(1) and GARCH(1,1) models, in the mean and in the variance part of the model, following a "from-the-general to the particular" specification modelling strategy. In general, we are driven to conclude that no robust evidence supports the presence of day effects. Admittedly, in some specifications, a negative

**Table 5. Tests of equality (mean, median, variance across the week days).**

| Test on (Null Hypothesis) | Method | $LP_{ETH}$ | $DLP_{ETH}$ |
|---|---|---|---|
| Equality of means | Anova $F$ stat | $F(6,364) = 0.032$ ($p = 0.999$) | $F(6,636) = 1.086$ ($p = 0.371$) |
| Equality of medians | Chi-squared stat | $\chi^2(6) = 4.730$ ($p = 0.579$) | $\chi^2(6) = 5.899$ ($p = 0.435$) |
| | Kruskal-Wallis | $\chi^2(6) = 6.993$ ($p = 0.322$) | $\chi^2(6) = 2.143$ ($p = 0.906$) |
| Equality of variances | Bartlett | $\chi^2(6) = 0.155$ ($p = 0.999$) | $\chi^2(6) = 3.137$ ($p = 0.792$) |
| | Levene | $F(6, 364) = 0.047$ ($p = 0.999$) | $F(6, 363) = 0.245$ ($p = 0.961$) |
| | Brown-Forsythe | $F(6, 1365) = 0.014$ ($p = 0.999$) | $F(6, 363) = 0.217$ ($p = 0.971$) |

Stats on $LP_{ETH}$ across the days of the week (53 obs)

| | Mean | Median | Std dev |
|---|---|---|---|
| Monday | 3.356 | 3.890 | 1.620 |
| Tuesday | 3.343 | 3.935 | 1.632 |
| Wednesday | 3.355 | 3.878 | 1.612 |
| Thursday | 3.374 | 4.088 | 1.617 |
| Friday | 3.364 | 3.934 | 1.630 |
| Saturday | 3.409 | 3.917 | 1.575 |
| Sunday | 3.423 | 3.919 | 1.570 |

Stats on $DLP_{ETH}$ across the days of the week (53 obs)

| | Mean | Median | Std dev |
|---|---|---|---|
| Monday | -0.025 | -0.025 | 0.158 |
| Tuesday | +0.037 | 0.037 | 0.190 |
| Wednesday | +0.013 | 0.019 | 0.164 |
| Thursday | +0.019 | -0.012 | 0.174 |
| Friday | -0.010 | 0.002 | 0.182 |
| Saturday | +0.044 | 0.003 | 0.159 |
| Sunday | +0.014 | -0.003 | 0.166 |

Note: The mean equality test is based on the analysis of variance (ANOVA), in which the null is that the data subgroups have the same mean. The median distribution equality is evaluated through two statistics: the chi-squared test–sometimes labelled as the median test (Conover, [26])–is an ANOVA test based on the comparison of the number of observations above and below the overall median in each subgroup; the Kruskal-Wallis test uses only ranks of the data; hence, it is less sensitive to outlier values. Three statistics are provided to assess the variance equality: in all cases, the null hypothesis is that the variances in all subgroups of data are equal, against the alternative that at least one subgroup has a different variance. The Bartlett test compares the log of the weighted average variance with the weighted sum of the log of variance; however, this test is sensitive to departure from normality distribution. The Levene test is based on an analysis of variance of the absolute difference from the mean, while the Brown-Forsythe is a variant of Levene's test, in which the absolute mean difference is substituted by the absolute median difference; the latter is recognised to be superior in terms of robustness and power–see Conover et al. [27].

Monday effect appears in the mean part of the model–which means that a significant lower daily return occurs on Monday, ceteris paribus–, and a negative and positive day effect in the variance part of the model–that is, lower and larger variance in daily returns–emerge for Saturday and Thursday, respectively. Table 6 reports some of the considered specifications, in which day effect are statistically significant. However, the statistical significance is not robust to slightly modified model specification, as Table 6 itself shows.

It is important to underline that in all cases, no Monday (or post-vacation) effect emerges in the variance part of the model. This is comprehensible: the BAYC NFT markets do not close in the weekends, and there is no reason to expect information over-flow on Monday (differently from the real no-crypto markets). Similarly, it is not surprising that weekend day effects

**Table 6. Evaluation of day effects.**

**Dependent variable:** $DLP_{ETH}$

**Sample:** May 3 2021-May 8 2022

| MODEL: | ARCH(4) | | GARCH(1,1) | |
|---|---|---|---|---|
| | Mean equation | | | |
| C | 0.020 (2.34)** | 0.018 (2.24)** | 0.023 (2.85)*** | 0.024 (2.92)*** |
| DUM_Monday | -0.048 (-1.82)* | -0.049 (-1.98)** | -0.052 (-2.73)*** | -0.052 (-2.63)*** |
| | Variance equation | | | |
| C | 0.024 (11.64)*** | 0.022 (10.65)*** | -0.001 (-1.60) | -0.0002 (-0.18) |
| RESID(-1)_squared | 0.165 (2.84)* | 0.188 (3.12)*** | 0.147 (4.55)*** | 0.157 (4.34)*** |
| RESID(-2)_squared | -0.018 (-0.66) | -0.016 (-0.66) | | |
| RESID(-3)_squared | -0.039 (-1.96)** | -0.040 (-2.36)** | | |
| RESID(-4)_squared | 0.064 (3.92)*** | 0.103 (4.91)*** | | |
| GARCH(-1) | | | 0.836 (23.75)*** | 0.814 (19.56)*** |
| DUM_Thursday | | -0.0004 (-0.09) | 0.013 (3.45)*** | 0.012 (3.00)*** |
| DUM_Saturday | -0.011 (-2.85)*** | -0.011 (-3.16)*** | | -0.002 (-0.55) |
| | Diagnostic | | | |
| S.E. of regression | 0.172 | 0.172 | 0.172 | 0172 |
| Log likelihood | 157.18 | 161.07 | 154.4 | 154.5 |
| Durbin-Watson stat | 2.57 | 2.57 | 2.57 | 2.57 |
| Akaike info criterion | -0.806 | -0.822 | -0.802 | -0.797 |
| Schwarz criterion | -0.722 | -0.726 | -0.739 | -0.723 |
| F-stat [p(F-stat)] | 0.432 [0.882] | 0.374 [0.934] | 0.575[0.719] | 0.031 [0.999] |

Note: $z$-stat in parenthesis; starred coefficient are significant at the 1% (***), 5% (**) or 10%(*) level. $F$-test is on the joint significance of day effects.

are absent in the mean part of the ARCH/GARCH model: there is no reason to believe that Saturday or Sunday are different from work days in the world of crypto activities. Admittedly, formal financial markets are closed over the weekend, but this fact does not seem to exert any effects on BAYC NFT prices.

A possible explanation for the negative Monday effect in the mean part of the model describing daily return–which is far from being a robust result, since it does not emerge from simple descriptive statistics, and show statistical significance only in some specifications in regression analysis–can be due to a supply-excess on Monday. As a matter of fact, Monday is the day in which the highest number of sales, on average, occurs, as documented in Table 7 (however, formal test cannot reject the null that the average values are constant across the seven days). A similar explanation, related to the number of sales, may be offered for the higher variance in daily returns on Thursday: Thursday is the day with the lowest number of sales, on

**Table 7. Number of sales across the days of the week.**

| | SALES_1 | SALES_2 | SALES_3 | SALES_4 | SALES_5 | SALES_6 | SALES_7 |
|---|---|---|---|---|---|---|---|
| Mean | 79.43 | 59.02 | 52.28 | 47.83 | 51.55 | 62.57 | 68.59 |
| Median | 24.0 | 20.0 | 21.0 | 24.0 | 28.0 | 26.0 | 25.0 |
| Maximum | 1120.0 | 513.0 | 388.0 | 287.0 | 320.0 | 440.0 | 588.0 |
| Minimum | 4.00 | 6.0 | 7.0 | 2.0 | 5.0 | 2.0 | 3.0 |
| Std. Dev. | 176.52 | 93.78 | 75.32 | 58.43 | 64.27 | 89.58 | 110.64 |

Note: _1 stays for Monday, _2 stays for Tuesday, etc..

average, so that, average price is easily influenced by single item prices, with a possible effect on larger variance, ceteris paribus. (If we considered the number of sales as a regressor in the mean part of the ARCH/GARCH model, it would assume a negative and significant coefficient, leading the Monday effect to become insignificant. However, information criteria suggest that the day effect is preferable to the number of sales, as a regressor, in the mean part of the model. The number of sales is never significant in the variance part of the model).

## Relations with financial markets: Equities and commodities

In order to evaluate whether daily BAYC NFT prices are related with financial market dynamics, we consider the daily observation of Standard & Poor's 500 index, denoted by *SP500* (its log is *LSP500* and the first difference *DLSP500*). Specifically, we consider the close-price; outcomes do not change, if open-prices are considered. (*SP500* index on Saturday, Sunday and national public holidays are considered equal to the previous day, to keep 7 observations per week in subsequent analysis; substantial results do not change if we consider 5 observations per week, dropping the weekend observations of the BAYC NFT prices). As it is well known in finance literature, also daily observations of *SP500* follow a I(1) statistical process. This holds also in the sample under present consideration (see Table 8). In the period at hand, we find here that $P_{ETH}$ and *SP500* show a low simple correlation (around 0.23), and are not co-integrated according to usual tests (*à la* Engle-Granger, or *à la* Johansen). The absence of co-integration means that no long-run link between variables is operative. Variables in first-differences are not correlated as well, with a very low simple correlation. No causality link is significant, in any direction, according to the Granger causality concept (up to 7 lags are considered, though Table 8 reports only the test results considering the two-lag case). These conclusions hold for the series both in level and in log (Tables report only the evidence concerning variables in log). The same substantial results emerge, if one considers the Dow Jones Industrial Average Index (*DJ*) instead of Standard and Poor's 500 (results on *DJ* are available form Authors upon request); this is not surprising, provided that the simple correlation between the daily values of these two indices is above 0.8 in the period under current investigation.

Again, the same results, that is, the lack of co-integration and low correlation between variables (both in levels and in first difference) emerges as far as the relations concern between the price of BAYC NFTs, on the one side, and the price of some specific goods of financial relevance, such as gold or crude oil, on the other side. The results concerning the relations between the daily prices of BAYC NFTs and daily price of gold are reported in Table 9.

These results are in line with the outcome of some contributions available in literature, showing that crypto activities (and specifically the exchange rates of cryptocurrencies *vs*. institutional currencies) show low correlation with institutional financial markets (see, e.g., Ciaian et al. [28]; Li and Wang [29]; Bouri et al. [30]; Virk [31]). Neither BAYC NFT prices show a high correlation with the series of the exchange rates of Ether (or Bitcoin) against Euro or US Dollar, in our databank. Also in these cases, the daily series of all these exchange rates are integrated of order 1, but no co-integration relation with $P_{ETH}$ emerges. Again, no Granger causality relations emerge, for variables in levels or in first difference, as far as $P_{ETH}$ and cryptocurrency exchange rates are concerned.

Also this set of results from our databank is in line with available evidence concerning other NFTs; specifically, Borri et al. [1] document that the majority of NFT market variation is not captured by cryptocurrency markets, even if the excess return of NFT market is associated with the excess return of the crypto-currencies used in the NFT exchanges. As already mentioned, Dowling [3] finds limited links between cryptocurrency and NFT markets (even if

**Table 8. Relation between $P_{ETH}$ and SP500.**

| | | Unit root test | |
|---|---|---|---|
| | | ADF | PP |
| | | $t$ ($p$-value) | $t$ ($p$-value) |
| *LSP500* | | -0.021 (0.407) | -1.896 (0.334) |
| *DLSP500* | | -15.89 (0.000)*** | -19.64 (0.000)*** |

**Cointegration between LPETH and LSP500**

**Engle-Granger procedure**

ADF on co-integration regression residuals

$t$ = -1.227 (asymptotic McKinnon $p$ = 0.202) (no constant, 1 lags)

$t$ = -1.77 (asymptotic McKinnon $p$ = 0.255) (no constant, 2 lags)

$t$ = -1.343 (asymptotic McKinnon $p$ = 0.166) (no constant, 7 lags)

**Johansen procedure**

| No. of CE(s) | Eigenvalue | Trace Stat/ Max Eigenv. | 0.05 Crit val | $p$-value |
|---|---|---|---|---|
| Trace test: | | | | |
| None | 0.024 | 10.963 | 15.49 | 0.214 |
| At most 1 | 0.006 | 2.139 | 3.841 | 0.144 |
| Max eigenvalue test | | | | |
| None | 0.024 | 8.823 | 14.26 | 0.321 |
| At most 1 | 0.006 | 2.139 | 3.841 | 0.144 |

**Correlation**

Corr ($LP_{ETH}$,*LSP500*) = 0.542

Corr ($DLP_{ETH}$,*DLSP500*) = -0.007

**Granger causality**

| H0: | $F$ | $p$-value |
|---|---|---|
| *LSP500* does not Granger-cause $LP_{ETH}$ | $F_{2,364}$ = 1.016 | 0.370 |
| $LP_{ETH}$ does not Granger- cause *LSP500* | $F_{2,364}$ = 0.147 | 0.866 |
| *DLSP500* does not Granger- cause $DLP_{ETH}$ | $F_{2,363}$ = 1.241 | 0.290 |
| $DLP_{ETH}$ does not Granger- cause *DLSP500* | $F_{2,363}$ = 0.560 | 0.571 |

Note: ADF and PP indicate, respectively, the augmented Dickey-Fuller and the Phillips-Perron test in which the null is the presence of a unit root in time series. Model selection, where appropriate and not differently specified, is based on the Schwarz information criterion. In PP test procedure, Bartlett kernel is used to select the spectral estimation method and Newey-West procedure is used to select the bandwidth. *,**,*** denote significant at the 10%, 5%, 1% level, respectively. Sample from May 3, 2021 to May 8, 2022 (371 obs.), adjusted according to considered lags: the number of observations, after adjustment, is 368 and 369 for *LSP500* and *DLSP500*, respectively. In the Engle-Granger co-integration test procedure, the information criteria lead to different lag order concerning the unit root test on residuals (Schwarz criterion selects 1 lag); asymptotic $p$-values *à la* McKinnon are reported. In the Johansen procedure, $p$-values of trace and eigenvalue test are *à la* McKinnon-Haug-Michelis. Granger causality is tested using 2 lags, with 369 and 368 observations for the relation among levels and first-differences, respectively.

non-parametric wavelet coherence analysis shows that co-movements between the two markets exists), and also limited correlation among different NFT items. On the opposite, Kong and Lin [4] observe that a NFT index based on *CryptoPunk* prices positively co-moves with the exchange rate of Ether against US Dollar, but the result is derived over the 5-year period 2017 to 2022: in this long period of time, all crypto markets have been growing, and the positive correlation is not surprising (in fact, the correlation is lower, if evaluated only over the last year). Kong and Lin [4] also find that the stock market in US (as proxied by NASDAQ index) has little impact on the prices of NFTs; the same holds for the stock markets of U.K., Germany, Japan, China, and Hong Kong (as measured by FTSE Index, DAX Index, Nikkei Index, SSE

**Table 9. Relation between $P_{ETH}$ and $P_{GOLD}$ (per-ounce price of gold, in USD).**

| | | Unit root test | |
|---|---|---|---|
| | | ADF | PP |
| | | $t$ ($p$-value) | $t$ ($p$-value) |
| $LP_{GOLD}$ | | -2.292 (0.175) | -2.196 (0.208) |
| $DLP_{GOLD}$ | | -20.02 (0.000)*** | -20.10 (0.000)*** |
| | | Co-integration between $LP_{ETH}$ and $LP_{GOLD}$ | |
| | | Engle-Granger procedure | |
| | | ADF on co-integration regression residuals | |
| $t$ = -2.055 (asymptotic McKinnon $p$ = 0.499) (no constant, 1 lags) | | | |
| $t$ = -2.190 (asymptotic McKinnon $p$ = 0.429) (no constant, 2 lags) | | | |
| $t$ = -2.568 (asymptotic McKinnon $p$ = 0.253) (no constant, 7 lags) | | | |

| | | Johansen procedure | | |
|---|---|---|---|---|
| No. of CE(s) | Eigenvalue | Trace Stat | 0.05 Crit val | $p$-value |
| Trace test: | | | | |
| None | 0.029 | 13.643 | 15.49 | 0.0933* |
| At most 1 | 0.008 | 3.042 | 3.841 | 0.0862* |
| Max eigenvalue test | | | | |
| None | 0.029 | 10.601 | 14.26 | 0.174 |
| At most 1 | 0.014 | 3.042 | 3.841 | 0.081* |

| *Correlation* | | |
|---|---|---|
| Corr ($LP_{ETH}$,$LP_{GOLD}$) = 0.131 | | |
| Corr ($DLP_{ETH}$,$D LP_{GOLD}$) = 0.015 | | |

| *Granger causality* | | |
|---|---|---|
| H0: | $F$ | $p$-value |
| $LP_{GOLD}$ does not Granger-cause $LP_{ETH}$ | $F_{2,364}$ = 0.677 | 0.509 |
| $LP_{ETH}$ does not Granger-cause $LP_{GOLD}$ | $F_{2,364}$ = 1.210 | 0.299 |
| $DLP_{GOLD}$ does not Granger-cause $DLP_{ETH}$ | $F_{3,360}$ = 1.446 | 0.229 |
| $DLP_{ETH}$ does not Granger-cause $DLP_{GOLD}$ | $F_{3,360}$ = 1.981 | 0.116 |

Note: Daily prices of gold–in US Dollar, per ounce–are from World Gold Council, www.gold.org; specifically, the close-price is considered; prices on Saturday, Sunday and holidays are set equal to the previous day. ADF and PP indicate, respectively, the augmented Dickey-Fuller and the Phillips-Perron test in which the null is the presence of a unit root in time series. Model selection, where appropriate and not differently specified, is based on the Schwarz information criterion. In PP test procedure, Bartlett kernel is used to select the spectral estimation method and Newey-West procedure is used to select the bandwith. *,**,*** denote significant at the 10%, 5%, 1% level, respectively. Sample from May 3, 2021 to May 8, 2022 (371 obs.), adjusted according to considered lags; the number of observations, after adjustment, is 368 for both $LP_{GOLD}$ and $DLP_{GOLD}$. In the Engle-Granger cointegration test procedure, the information criteria lead to different lag order concerning the unit root test on residuals (Schwarz criterion selects 1 lag); asymptotic $p$-values à la McKinnon are reported. In the Johansen procedure, $p$-values of trace and eigenvalue test are à la McKinnon-Haug-Michelis. Granger causality is tested using 2 or 3 lags, with 369 and 367 observations for the relation among levels and first-differences, respectively.

Index, and Hang Seng Index, respectively). At most, a positive association in terms of correlation may be found between the returns of stock markets and the returns from NFTs (again, over the long period of 5-yeat time span 2017–22). It is worth reporting that Ante [32, 33], resorting to VAR models and co-integration analysis, finds that Bitcoin and Ether exchange rate dynamics affect NFT volume of transaction and the number of wallets that trade NFTs; more openly stated: according to Ante's analyses, co-integration relation exists, with causality running from cryptocurrency value to the volume of NFT transaction; however, Ante does not consider prices or price indices.

Thus, the appropriate conclusion to draw is that $P_{ETH}$ behaves as the price of a traditional financial asset (like equity and currency) and as financial indices (such as Dow Jones Industrial

Average or Standard & Poor's 500)–in the sense that the daily series are integrated of order 1, highly volatile and with autoregressive heteroscedasticity; however, the simple correlation between $P_{ETH}$ and such financial variables is low. On the one side, this piece of evidence says that no contagion phenomena occur, from the standard financial markets to NFT market, and this may suggest that different agents are operative in the two markets. On the other side, this feature makes BAYC NFTs appropriate tools for financial portfolio diversification, according to possibly different goals and strategies. More in general, crypto-currencies and assets in crypto-markets can be good diversifiers in financial portfolios, due to their low correlation with traditional assets. In this respect, it is worth mentioning the study of Živkov et al. [34] concerning the optimal diversification of portfolio with investment in Bitcoin; in particular– also resorting to GARCH models–they study the combination between Bitcoins and traditional assets (including *SP500*, gold, crude oil) that entails the best downside risk-minimizing performances of portfolio. Of course, similar exercises can be repeated with reference to BAYC NFTs; preliminary evidence suggest that the best downside risk minimization can be obtained combining BAYC NFTs with *SP500*; however, specific and deeper investigations in this regard are left to future research.

## Relations with the exchange rate: Euro and Dollar

The daily series of the ETH-EUR exchange rate is integrated of order 1, and not co-integrated with $P_{ETH}$. The same holds for the exchange rate ETH-USD. Thus, it is interesting to evaluate whether the dynamics of the prices of BAYC NFTs maintain the same pattern and properties, if expressed in EUR or USD instead of ETH. The point is far from being obvious, since people likely take their decision concerning BAYC NFTs, basing on price in traditional currency, which is used in everyday life, rather than in cryptocurrency (the point that people take decisions on financial portfolio composition basing on price and return expressed in domestic currency is made, among others, by Fidora et al. [35]). However, we find that the results associated to the prices of BAYC NFTs translated in EUR or USD are fully in line with the outcomes from BAYC NFT prices in ETH. As a matter of fact, the simple correlation between the series of BAYC NFT prices, expressed in ETH and in EUR, is 0.972 (and 0.923 if considered in log); the correlation between price expressed in ETH and in USD is 0.968 (and 0.918 if considered in log), Thus, it is far from being surprising that the substantial results are the same, irrespective of the fact that prices are considered in ETH, EUR or USD. It is worth reporting that the coefficient of variation (i.e., the relative standard deviation) of the daily time series of average prices, is 0.76, 0.73 and 0.71, if prices are expressed in ETH, EUR and UDS, respectively. Thus, the variability of the daily series slightly decreases, if prices are considered in "official" currencies, instead of cryptocurrency.

However, significant changes in the exchange rate have occurred in the time period under consideration: the Ether followed an appreciation pattern against Euro and US Dollar over July-October 2021, while it has been following a devaluation pattern since November 2021, with respect to both Euro and US Dollar -See Fig 2. (See Chu et al. [36] and Virk [31], among others, for further investigation on the distribution and properties of cryptocurrencies' exchange rates. Incidentally, the devaluation pattern of ETH, with respect to both Euro and US dollar, has been continuing also after the time of the sample under scrutiny, between August and December 2022).

As already mentioned, available studies document a limited impact of cryptocurrency dynamics on NFT prices (see, e.g., Borri [1] and Dowling [3]). We confirm such conclusions also with reference to the case of BAYC NFTs.

According to the evidence from our present dataset, in a first sub-period–namely, May-October 2021, when the growth rate of BAYC NFT price in ETH is much larger as compared

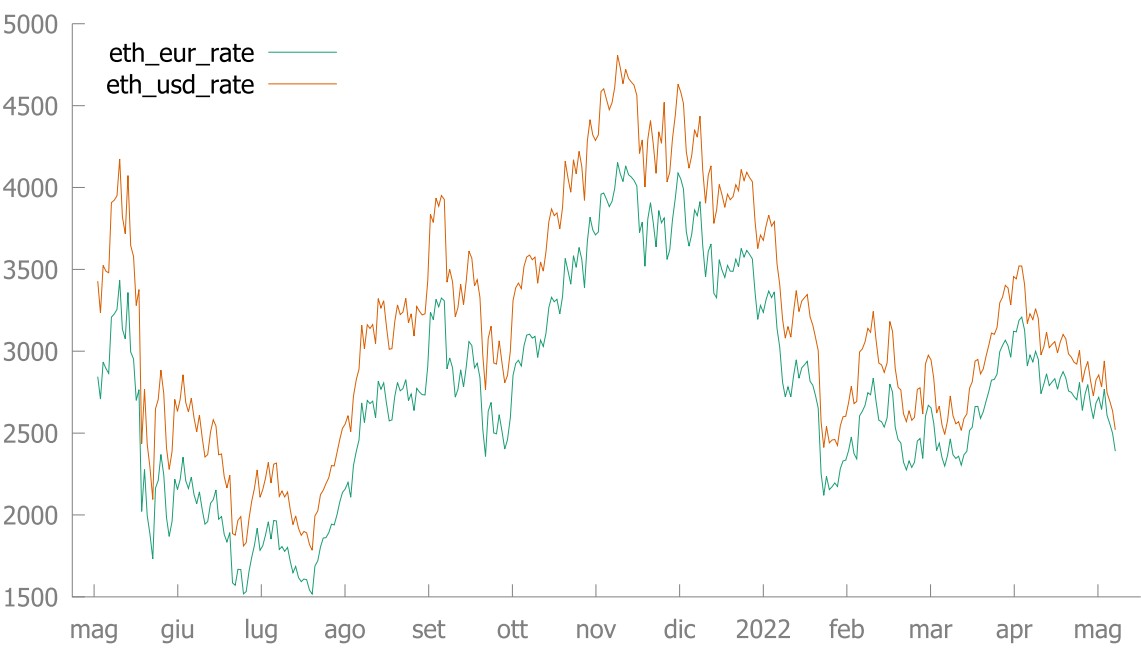

**Fig 2. Exchange rate of Ether w.r.t. Euro and USD (May 3, 2021—May 8, 2022).**

to the subsequent sub-period–the correlation between the exchange rate of Ether (against Euro and US Dollar) and BAYC NFT price (in ETH) is positive and significant, while it is negative and significant in the second sub-period. Thus, it is not surprising that the correlation is not significant if evaluated over the whole time span under scrutiny (such a result is in line with available results from existing literature). However, it is interesting to underline that, from November 2021 to May 2022, daily average prices have been moving in the opposite direction of the exchange rate of ETH: when ETH depreciates, prices of BAYC NFT tend to increase; when ETH appreciates, prices tend to decrease. At the moment, we can not state whether this is a (new) structural tendency or a transitory tendency. As mentioned above, in the weeks after the time period under consideration in the formal analysis of this paper, ETH has been devaluating against official currencies (USD and EUR) and BAYC NFT prices in ETH have been decreasing. Thus, at the moment, a wise conclusion seems to be that the correlation between BAYC NFT prices and cryptocurrency exchange rate is not significant, if evaluated over a 1-year period of time.

## 5. Additional analysis: A price index from a hedonic approach

Since data are available for every collectible piece of BAYC collection exchanged during the period under scrutiny, we can resort to the hedonic price regression technique (see Ginsburgh et al. [37]) to obtain a price index, which controls for the features of each traded piece of the collection. The outcome may also represent a robustness check of the results obtained in the previous sections. Specifically, we can run the regression

$$\ln(p_{it}) = \sum_t \sum_j \theta_{jt} w_{ijt} + \sum_{t=1}^{T} \delta_t c_{it} + \varepsilon_{it}, \tag{3}$$

where BAYC NFT collectibles are indexed by $i$, time periods (i.e, days) by $t$ and characteristics by $j$; $p$ is sales price, $w$'s are characteristics, $c$'s are time period dummies (daily frequency).

**Table 10. GARCH models for DLP$_{IEDH}$.**

**Dependent variable:** $DLP_{IEDH}$

***Sample*: 03/5/2021-08/5/2022 (366 obs after adjustments)**

| MODEL: | GARCH(1,1) | ARCH(4) | ARCH(1) | ARCH(1) |
|---|---|---|---|---|
| | | Mean equation | | |
| C | -0.006 (-0.63) | -0,006 (-0.61) | -0.005 (-0.58) | 0.0002 (0.58) |
| DUM_Monday | | | | -0.06 (-2.79)*** |
| | | Variance equation | | |
| C | 0.022 (14.59)*** | 0.020 (22.09)*** | 0.020 (22.58)*** | 0.016 (11.78)*** |
| RESID(-1)_squared | 0.308 (4.12)*** | 0.33 (4.12)*** | 0.314 (4.16)*** | 0.452 (4.22)** |
| RESID(-2)_squared | | 0.009 (0.25) | | |
| RESID(-3)_squared | | -0.03 (-1.03) | | |
| RESID(-4)_squared | | 0.02 (1.17) | | |
| GARCH(-1) | -0.095 (-1.88)* | | | |
| DUM_Thursday | | | | 0.017 (6.38)*** |
| | | Diagnostic | | |
| S.E. of regression | 0.179 | 0.179 | 0.178 | 0.178 |
| Log likelihood | 160.0 | 160.9 | 158.9 | 167.7 |
| Durbin-Watson stat | 2.65 | 2.66 | 2.66 | 2.63 |
| Akaike info criterion | -0.853 | -0.846 | -0.852 | -0.889 |
| Schwarz criterion | -0.810 | -0.783 | -0.820 | -0.836 |

Note: *z*-stat in parenthesis; starred coefficient are significant at the 1% (***), 5% (**) or 10% (*) level.

As far as the characteristics concern, every BAYC NFT is described through 104 dummy variables capturing the presence or absence of specific features, referring to, e.g., the color of the short, the hat, the facial expression, and so on. Coefficients of the characteristics are statistically significant and their analysis will be provided in a companion paper.

The coefficients of interest in the present investigation are the δ-coefficients. The value of the index $\pi_t \equiv \exp(\hat{\delta}_t)$ describes the price of a characteristics-free commodity, that is, the price of a collectible, after having controlled for its characteristics. The series, denoted as $P_{IHED}$ (where subscript IHED is intended for 'index, hedonic'), is computed to assume value equal to 1 for the first available observation; such series is integrated of order 1 and its correlation with $P_{ETH}$ is 0.975 (or 0.995 if variables are in log-values). Also in this case, hence, there is no surprise from the fact that this series shares all the properties with the series given by the simple average of the prices of daily sales. This also means that prices are mainly driven by the (stochastic) time trend, and the characteristics of the pieces, though statistically significant, play a minor role in shaping price, and notably price dynamics. Table 10 reports the estimation of some specifications of the ARCH/GARCH model for the first difference of the log of $P_{IHED}$. The ARCH(1) specification appears to be the best one, according to the Schwarz criterion. Thus, one can conclude that the error variability, and its persistence, are lower when evaluated with reference to the series of the price of characteristics-free collectibles. In other words, the effect of the characteristics upon the price of collectibles entails an increase of the price variability and its persistence–which support the point that characteristics do matter. The day of the week effects maintain the same properties as in the case of the dynamics of daily simple average price: in some specifications, Monday has a negative effect on the daily return in the mean part of the model, while the variance of returns tends to be larger for the sales occurring on Thursday.

## 6. Conclusions

This paper has provided an analysis of the price dynamics of a set of non-fungible tokens (NFT), specifically the 10,000 pieces of the BAYC collection, sold–through cryptocurrencies–mainly on the OpenSea platform. This collection was released in Spring 2021; over the first 12 months of life, average prices have multiplied by about 200. One can wonder whether these tokens are "art" pieces or simply financial assets. Buyers' motivations should be investigated. Collectors are active in this market, and the sake for collection may explain part of the phenomenon. Belonging to an "exclusive" club may provide social recognition and intrinsic utility. NFTs are born exactly to make *rare* what can be infinitely duplicated.

We are aware that several other potential aspects of price determinants have been overlooked in the present analysis; for instance, we have not taken into account the interaction with the markets of other NFT items or the effect of network among subjects trading NFTs, which could potentially play a role. For sure, NFT markets and the scientific (economic, financial, but also sociological) research on them are at their early stages, so that consensus is not established, on the truly relevant determinants, and the best perspective of analysis to take.

Our present analysis has shown that the BAYC NFT prices (measured in different ways) share basic characteristics with the prices of financial assets. Notably, they are non-stationarity and the daily return rates show heteroscedasticity and autoregressive variance. Thus, ARCH/GARCH models can be used to describe the dynamics of such prices, in a time series analysis perspective. Our analysis has also shown that there is limited evidence of significant day effects, in both the mean part and the variance part of the model. A negative Monday effect in the mean part of the model appears to be significant in some specifications, but the BAYC NFT prices do not show larger variance after weekends. This could be also due to the fact that crypto markets do not close in the week-end, so there is no information over-flow following closure day. Incidentally, day effects loose importance, in recent time, also for traditional financial assets.

Our conclusion is that BAYC NFT share relevant features with financial assets, and, at the same time, the characteristics of the collectibles–which have to do with their aesthetic/artistic content–do matter, as long as they are statistically significant in explaining the individual price levels of each specific collectible. However, the largest part of price levels is driven by the stochastic trend, with typical features of financial assets.

Interestingly, the price series of these non-fungible tokens is related neither with the series of assets and commodities of the "real" world, nor with the exchange rates of cryptocurrencies. No contagion phenomena from the standard financial market to the BAYC NFT appears to occur. This feature–one can suggest–makes these non-fungible tokens optimal tools to diversify financial portfolios, even if the risk of bubble burst is far from being absent, and the fundamentals of BAYC collectibles prices are difficult to define and to measure.

## Supporting information

**S1 Data. Contains all data used in the analysis (371 daily observations, over the sample 3 May 2021 to 8 May 2022); variables' names correspond to the names used in the article.** (XLSX)

## Acknowledgments

Although we alone are responsible for what is written here, we would like to thank the Editor and three anonymous referees for their valuable comments. Thanks also go to Roberto Di

Mari, Andrej Sakar, along with the participants to different university seminars and workshops, including EWACE 2022, for useful discussion.

## Author Contributions

**Conceptualization:** Roberto Cellini, Tiziana Cuccia.

**Data curation:** Johan Lyrvall.

**Formal analysis:** Johan Lyrvall.

**Investigation:** Johan Lyrvall.

**Methodology:** Roberto Cellini.

**Supervision:** Tiziana Cuccia.

**Writing – original draft:** Tiziana Cuccia.

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
