## [Decision Letter · Decision Letter 0]

8 Dec 2022

PONE-D-22-31290On the price dynamics of non-fungible tokens: The ‘Bored Apes’ casePLOS ONE

Dear Dr. Cellini,

Thank you for submitting your manuscript to PLOS ONE. After careful consideration, we feel that it has merit but does not fully meet PLOS ONE’s publication criteria as it currently stands. Therefore, we invite you to submit a revised version of the manuscript that addresses the points raised during the review process.

The paper merits major improvements in, primarily, three aspects:

a) motivation: NFTs are an exciting innovation of recent times, but the relationship of this uniqueness of their nature with the study of their price dynamics does not come across in good depth/detail as a topic of exploration

b) discussion of results: the economic interpretation of results merits more depth

c) the contributions of the study must be accentuated

Both reviewers have offered ample comments in that respect, alongside other issues, and I suggest the authors follow them.

We look forward to receiving your revised manuscript.

Kind regards,

Vasileios Kallinterakis

Academic Editor

PLOS ONE

Additional Editor Comments:

The paper merits major improvements in, primarily, three aspects:

a) motivation: NFTs are an exciting innovation of recent times, but the relationship of this uniqueness of their nature with the study of their price dynamics does not come across in good depth/detail as a topic of exploration

b) discussion of results: the economic interpretation of results merits more depth

c) the contributions of the study must be accentuated

Both reviewers have offered ample comments in that respect, alongside other issues, and I suggest the authors follow them.

Reviewers' comments:

Reviewer's Responses to Questions

**Comments to the Author**

1. Is the manuscript technically sound, and do the data support the conclusions?

Reviewer #1: No

Reviewer #2: Partly

2. Has the statistical analysis been performed appropriately and rigorously? 

Reviewer #1: No

Reviewer #2: N/A

3. Have the authors made all data underlying the findings in their manuscript fully available?

Reviewer #1: No

Reviewer #2: Yes

4. Is the manuscript presented in an intelligible fashion and written in standard English?

Reviewer #1: No

Reviewer #2: No

5. Review Comments to the Author

Reviewer #1: I thank you for your time and for submitting this paper. However, you need to rethink your research scope, research design, appropriateness of methodology and why we should care about “Bored Apes Yacht Club”?

I started reading and I was very much interested in the contribution the research may bring into the area of investigating NFTs collections’ price dynamics, volatility and/ or investors behaviour.

However, I realised after the first page, that the submitted paper is a free and wild wonder in the realm of NFTs. The paper mentions in page 3 that it has a “core” question: “Do NFTs represent a new medium of artistic expression in the Crypto art field or a financial bubble in the making?” First, this is not one question. These are two distinct questions. The short answer for the first question; yes, it is a new medium, you clearly stated that it was launched in April 2021. I was not sure how you would measure the newness of an artistic expression. Anyway, the research paper ditched this part anyway in the following sections. Second, the paper did not address the second question.

1- You stated that “Our results show that the “artistic characteristics” do matter, but, at the same time, the price dynamics share relevant features with the typical dynamics of a pure financial asset.”

Comment:

- How artistic characteristics matter? In what sense? There are no defined characteristics you investigated and examined the interrelationship with any other financial NFT variable.

- What is a “pure” financial asset?

2- Page 4: You stated “Day effects in daily return –which, to the best of our knowledge, are not yet studied by available literature on NFT– emerge to play a limited role, in both the mean and the variance part of the model; this evidence is common to financial assets in recent years”

Comment:

- You did not have to run an analysis on NFT time series to find that. You clearly stated in page 9, “This is comprehensible: the BAYC NFT markets do not close in the weekends, and there is no reason to expect information overflow on Monday (differently from the real no-crypto markets). Similarly, it is not surprising that weekend day effects are absent in the mean part of the ARCH/GARCH”

- which financial assets are you comparing NFTs with? And why?

3- Page 12: It is not clear which asset you are referring to here: Price of Ethereum behaves as the price of a financial asset, but it is not correlated with financial variables capturing the daily dynamics of financial markets.

4- Page 12 You concluded that the absence of contagion between markets (“standard”? market and “NFT?” market, can lead to a better portfolio diversification. There are portfolio optimisation methods which can help you come up with this conclusion. “piece of evidence says that no contagion phenomena occur, from the standard financial markets to NFT market, and this may suggest that different agents are operative in the two markets. On the other side, this feature makes BAYC NFTs optimal financial tools for financial portfolio diversification”

5- Rethink the use of language: Page 12 ("roughly speaking” the Ether followed an appreciation pattern against Euro and US Dollar over July-October 2021). It either followed or it did not.

6- Page 13: you moved to investigating price movement patterns between BAYC NFT and Euro and US dollar. The rationale is missing.

Reviewer #2: Explanation of the answers given above:

1. Is the manuscript technically sound, and do the data support the conclusions?

Please, use words and not symbols in the text, where appropriate. E.g. “We consider the residuals from the OLS regression of DLPETH on the constant term, and test for a autoregressive process concerning the squared residuals.” Use words for variable DLPETH.

2. Has the statistical analysis been performed appropriately and rigorously?

The answer to this question is: “Partly”, but this option is not offered in the reviewer form.

I suggest using more advanced GARCH models (such as DCC GARCH), calculation of dynamic correlations between DJIA index/gold/crude oil and NFT, finding out downside risk of portfolios combined of NFT, on one hand, and DJIA, crude oil and NFT, on the other hand, by using parametric and semiparametric Value-at-Risk and Conditional Value-at-Risk (see the paper Živkov et al. (2021), Measuring Downside Risk in Portfolios with Bitcoin).

4. Is the manuscript presented in an intelligible fashion and written in standard English?

The professional proofreading of the text must be done by the native English speaker.

Please, try to avoid clichés like “last but not least”.

Major comments:

1. There are lots of statistical analysis but economic intuition and interpretation are missed at several places.

2. Behavioral explanations of NFT price patterns must be strengthened.

3. Comment on the intrinsic value of NFT.

4. Comment on how the sample size affect the results.

5. What is the main contribution of the paper to the existing literature?

6. Why did you choose DJIA and not S&P 500 or some other index?

Minor comments:

1. Conclusion must not contain new evidence and new literature review not mentioned previously.

2. Please, explain the findings of Kong and Lin (2021).

3. Table 1 – Instead of “Basic statistics” use “Descriptive Statistics”. There is no need to explain what is skewness and kurtosis in the Notes.

4. In Note (for table 9) you wrote “See Note to Table 2 for the unit root tests”. Please, repeat what is written in Note to the Table 2 about unit root tests in notes to all tables, where appropriate.

5. The correct name for the statistical software is EViews.

6. Replace “differentiate financial portfolios” by “diversify financial portfolios”.

6. PLOS authors have the option to publish the peer review history of their article (what does this mean?). If published, this will include your full peer review and any attached files.

Reviewer #1: No

Reviewer #2: No

---

## [Author Response · Author response to Decision Letter 0]

7 Apr 2023

A Letter explaining how the criticisms of the Reviewers and the Editor have been met, is up-load. Please, see the Letter (Here we copy its content).

= = = = =

First of all, let us thank the Reviewers and the Editor for their very careful reading of this article and valuable comments. We also thank them for the encouraging evaluations.

Specifically, the Editor –following the points made by all Referees– asked for a deeper discussion on motivation, a deeper discussion of results and their economic interpretation, and a better specification of the contributions of the study.

Introduction, Conclusions and several passages in the main body of the article have been changed to meet these points. In this Letter we explain the changes in detail, following the specific points made by the Referees (The points of the Referees are summarised and reported in italics).

Reviewer #1: 

1. Motivation and research design - The paper mentions in page 3 that it has a “core” question: “Do NFTs represent a new medium of artistic expression in the Crypto art field or a financial bubble in the making?” First, this is not one question. These are two distinct questions. The short answer for the first question is yes. Second, the paper did not address the second question. - You stated that “Our results show that the “artistic characteristics” do matter, but, at the same time, the price dynamics share relevant features with the typical dynamics of a pure financial asset.” Comment: How artistic characteristics matter? In what sense? There are no defined characteristics you investigated and examined the interrelationship with any other financial NFT variable.- What is a “pure” financial asset?

The Referee was right. We have completely re-written the paragraphs containing the motivation of the research, and its contribution to the literature (see the paragraphs at the end of the Introduction, pp. 4 and 5 of the revised version). Some other passages in the main body of the paper have been rephrased, as well, to make the contribution of this paper clearer . We have also replaced the expression “artistic characteristics” with “traits of the images –like hair colour, eyes, clothes, which are elements of differentiation that can be also appreciated from an artistic/aesthetic point of view–…” Again, we have cut the adjective “pure” from financial assets (What we meant was “a standard financial asset”, but we agree that any adjective referred to financial assets may be simply confusing). We have also specified which financial assets we are referring to; specifically, we are referring to the price of equities, to the main indices of equity prices (specifically, DowJones and S&P500), to some raw commodities (specifically, crude oil and gold), and to exchange rates. We hope that such changes help avoiding mis-understandings. 

2. You stated “Day effects in daily return emerge to play a limited role, in both the mean and the variance part of the model; this evidence is common to financial assets in recent years”- Comment: You did not have to run an analysis on NFT time series to find that. You clearly stated in page 9, “This is comprehensible: the BAYC NFT markets do not close in the weekends, and there is no reason to expect information overflow on Monday (differently from the real no-crypto markets).. - Which financial assets are you comparing NFTs with? And why?

The Referee is right: The absence of significant day effects is not a core result of our analysis (and we put less emphasis on it in the revised version of the paper). In any case, a check of the possible presence of day effect was, in our opinion, necessary. Where appropriate, we specify which financial assets are under consideration in our comparisons (see below). 

3. Page 12: It is not clear which asset you are referring to here: Price of Ethereum behaves as the price of a financial asset, but it is not correlated with financial variables capturing the daily dynamics of financial markets.

In the revised version of the paper (at p. 13), we have explicitly stated which assets we are referring to –that is, traditional financial assets like equity and currency and financial indices such as DJ and SP500.

4. Page 12 - You concluded that the absence of contagion between markets (“standard”? market and “NFT?” market), can lead to a better portfolio diversification. There are portfolio optimisation methods which can help you come up with this conclusion. “Piece of evidence says that no contagion phenomena occur, from the standard financial markets to NFT market, and this may suggest that different agents are operative in the two markets. On the other side, this feature makes BAYC NFTs optimal financial tools for financial portfolio diversification”

We have re-written the sentences, and we have mentioned that NFT could be appropriate for portfolio diversification, according to possibly different strategies (p. 14 of the revised version). Even if portfolio optimization methods are clearly out of the goal of the present research, we have introduced a reference to a recent paper (Zivkov et al. 2021), specifically focused on the best portfolio diversification using Bitcoin [The reference to this paper was suggested by Referee #2 - see below]. Of course, the same exercise presented by Zivkov et al (2019) with respect to Bitcoins, could be replicated in our present paper with respect to BAYC NFT. We have not developed the exercise within our present paper, but we have mentioned the possibility. (By the way, preliminary evidence from such exercises suggest that –among the portfolios combined of NFT, on one hand, and SP500 / DJ / crude oil or gold, on the other hand– the best downside risk is associated with SP500, under the assumption of normal distribution; results change dramatically, like in Zivkov et al (2021), under different assumptions regarding return distribution, and conclusions seems to be not robust. However, we have preferred not to mention the outcome from such explorative exercise in the present paper: the exercise would deserve a different paper!) 

5. Rethink the use of language: Page 12 ("roughly speaking” the Ether followed an appreciation pattern against Euro and US Dollar over July-October 2021); It either followed or it did not.

Colloquial expressions have been avoided. A language revision by a professional society [LangageEdit.com] has been carried out. 

6. Page 13: you moved to investigating price movement patterns between BAYC NFT and Euro and US dollar. The rational is missing.

We explain the reason why we believe that it is worth investigating the dynamics of price in EUR or USD at the beginning of Section 5.2 of the revised version. In fact, the exchange ETH-EUR is integrated of order 1, and not co-integrated with PETH. The same holds for the exchange rate ETH-USD. Thus, it is interesting to evaluate whether the dynamics of the prices of BAYC NFTs maintain the same pattern and properties if expressed in EUR or USD, instead of ETH. The point is far from being obvious, since people may take their decision concerning demand and supply of BAYC NFTs, basing on price in traditional currencies, which is used in everyday life, rather than in cryptocurrency. However, as a matter of fact, no significant differences in outcomes emerges, if prices are considered in ETH or official (i.e., not crypto) currencies. In any case, the check is –in our opinion– necessary. 

Reviewer #2: 

General evaluation

- Please, use words and not symbols in the text, where appropriate. E.g. “We consider the residuals from the OLS regression of DLPETH on the constant term, and test for a autoregressive process concerning the squared residuals.” Use words for variable DLPETH.

We have substituted words to symbols where appropriate throughout the whole paper. 

- Has the statistical analysis been performed appropriately and rigorously? - The answer to this question is: “Partly”. I suggest using more advanced GARCH models [..](such as DCC GARCH for calculation of dynamic correlations between DJIA index/gold/crude oil and NFT, finding out downside risk of portfolios combined of NFT, on one hand, and DJIA, crude oil and NFT, on the other hand, [..](see the paper Živkov et al. (2021), Measuring Downside Risk in Portfolios with Bitcoin).

We have openly stated, in the main body of the revised version of the paper (Section 3) that we present only the results from “standard” GARCH models, and we do not deal with the problem of which variant of GARCH model could be the best, because –in the case at hand– variants like EGARCH or PARCH do not provide substantially different evidence, and there is no univocal compelling answer to the question about which is the best variant of the GARCH model for the case under scrutiny.

As to the dynamic correlation between BAYC NFTs and “standard assets”, evaluated via DCC GARCH, in order to derive the optimal portfolio composition, we would like to repeat here our thanks to Referee 2 for the suggestion of the reference of the article by Živkov et al. (we have to admit that we did not know this article). As already said in the Answer to point 4 of Referee 1, we have re-written the paragraph concerning portfolio composition, and we have mentioned that NFT could be appropriate for portfolio diversification, according to possibly different strategies. We have introduced the reference to Zivkov et al. (2021). Of course, the same exercise presented by Zivkov et al (2019) with respect to Bitcoins, can be replicated with respect to BAYC NFT. We have not developed the exercise within our present paper, but we have mentioned the possibility. (If one repeats the exercise of Živkov in the case of BAYC NFTs, preliminary results lead to the outcome that the portfolio with S&P500 index –among S&P500, DJ, crude oil and gold– has the best risk-minimizing results. However, like in Živkov, the result is highly sensitive to the assumption concerning the error distribution. Of course, a proper treatment of this point, in a scientific paper, should require a deep investigation, that goes beyond the scope of the present paper, and can be left for future research (and for future, specific paper). For this reason, in our present paper, we have preferred to mention the point, but without mentioning results from tentative exercises.

- The professional proofreading of the text must be done by the native English speaker.

Please, try to avoid clichés like “last but not least”.

As already mentioned, colloquial expressions have been removed. A language revision has been carried out by a professional society 

Major comments:

1. There are lots of statistical analysis but economic intuition and interpretation are missed at several places. 

and 

2. Behavioral explanations of NFT price patterns must be strengthened.

The economic meaning of the statistical evidence has been openly explained in several passages. E.g., the meaning of integration/stationarity of the time series (in Section 2, p. 7); the meaning of autoregressive heteroscedasticity (in Section 2, p. 8); the meaning of the possible day effect of Monday in the mean- and in the variance- part of the model (in Section 3, p. 11); the meaning of (no)-cointegration (in Section 5.1).

3. Comment on the intrinsic value of NFT.

A comment on the intrinsic value of NFTs has introduced at p. 4 (in Introduction). Given the multi-facet nature of NFT, it does not exist any univocal definition (and measure) of intrinsic value of NFTs, and it is problematic to propose a definition capturing all aspects of NFTs with artistic content.

4. Comment on how the sample size affect the results.

The size can be referred to both: (i) the number of transaction (per day), and (ii) the time dimension of the sample. 

As far as the point (i) concerns, we acknowledge that the number of transaction greatly varies across days –even if formal statistical tests do not sign statistical differences across the days of the week; on average, Monday is the day of the week with the highest number of transactions, and the possibility that this fact is the reason why a Monday effect emerges in some specifications is openly discussed in the revised version of the paper (See p. 10-11); 

As far as point (ii) concerns, we openly state that our results are stable over time (and stable across different time spans of the sample), with one exception --namely, the relation between the price of collectibles and the exchange rate of ETH. Indeed, there is no significant correlation, if evaluated over the whole time-span under consideration (May 3, 2021-May 8, 2022), while significant correlation (positive and negative, respectively) emerges if the sample is divided in to separate sub-sample. This outcome is openly stated, and commented also with reference to its economic implications (in Section 5.2, p. 15).

5. What is the main contribution of the paper to the existing literature?

We have clearly stated what is the main contribution of the paper to the existing literature in Introduction (p. 4-6 of the version with track changes).

In particular, the revised version of the paper reads: 

Our present analysis expands the studies on NFTs, focussing on daily prices of BAYC NFTs, […]. It contributes to the literature in two ways. 

First, the present study sheds light on the nature of non-fungible tokens with aesthetic contents, such as the BAYC collectibles. We show that aesthetic features of the collectibles do matter in the price determination; at the same time, price pattern share many characteristics with the pattern of financial assets like equities, stock indices and exchange rates. Specifically, […] Thus, these items are both collectibles and unique compositions that represent a new medium for artistic expression within the online “crypto art” field. In the meantime, they are investment assets and the price dynamics share relevant features with the price dynamics of financial assets. […]

Second, our study highlights specific characteristics of the daily price index of BATC NFTs, within a time series analysis framework. Under this perspective, our outcomes are largely in line with the evidence available for other NFTs. In particular, daily prices are non-stationary […]; the daily returns are highly volatile and show a heteroscedastic variance with an autoregressive pattern. […] Moreover, day effects in daily return –which, to the best of our knowledge, are not yet studied by available literature on NFT– emerge to play a limited role, in both the mean and the variance part of the model […] Finally, over the one-year period of time under investigation, the correlation of BAYC NFT returns with other assets in both the crypto- and the traditional (i.e., not crypto-) markets is low. 

Other specific contributions (e.g., on the relation of NFT prices with the exchange rate of crypto-currency) are underlined in the body of the article. 

6. Why did you choose DJIA and not S&P 500 or some other index?

There was no specific reason to choose DJ instead of SP500. However, since SP500 is a wider index than DJ, we have accepted the point of the referee, and we have reported the results referred to SP500 instead of DJ. Substantial results are unchanged. In fact, the correlation between DJ and SP500, in the time interval under present consideration (May 3,2021 to May 8, 2022) is 0.86; thus, it is not surprising that the results are substantially identical, irrespective of considering DJ or SP500. We openly recognize it in text. 

Minor comments:

1. Conclusion must not contain new evidence and new literature review not mentioned previously.

The Referee is right. We have modified some passages, in the main body of the paper and in the Conclusions, to meet the (correct) point. Now the concluding section does not contain any reference to previously un-mentioned literature contributions. The relevant contents of all articles mentioned in Conclusions have been presented and discussed in previous sections.

2. Please, explain the findings of Kong and Lin (2021).

Done (in Section 2) and briefly mentioned also in Conclusions.

3. Table 1 – Instead of “Basic statistics” use “Descriptive Statistics”. There is no need to explain what is skewness and kurtosis in the Notes.

Done (both points).

4. In Note (for table 9) you wrote “See Note to Table 2 for the unit root tests”. Please, repeat what is written in Note to the Table 2 about unit root tests in notes to all tables, where appropriate.

Done (in Table 8 and Table 9 of the revised version).

5. The correct name for the statistical software is EViews.

Correction done. Thanks.

6. Replace “differentiate financial portfolios” by “diversify financial portfolios”.

Done. Thanks.

= = = =

---

## [Decision Letter · Decision Letter 1]

2 May 2023

PONE-D-22-31290R1On the price dynamics of non-fungible tokens: The ‘Bored Apes’ casePLOS ONE

Dear Dr. Cellini,

Thank you for submitting your manuscript to PLOS ONE. After careful consideration, we feel that it has merit but does not fully meet PLOS ONE’s publication criteria as it currently stands. Therefore, we invite you to submit a revised version of the manuscript that addresses the points raised during the review process.

One reviewer has raised issues of general exposition, discussion and presentation of your paper and I suggest you heed those. 

We look forward to receiving your revised manuscript.

Kind regards,

Vasileios Kallinterakis

Academic Editor

PLOS ONE

Journal Requirements:

Additional Editor Comments:

One reviewer has raised issues of general exposition, discussion and presentation of your paper and I suggest you heed those.

Reviewers' comments:

Reviewer's Responses to Questions

**Comments to the Author**

1. If the authors have adequately addressed your comments raised in a previous round of review and you feel that this manuscript is now acceptable for publication, you may indicate that here to bypass the “Comments to the Author” section, enter your conflict of interest statement in the “Confidential to Editor” section, and submit your "Accept" recommendation.

Reviewer #2: All comments have been addressed

Reviewer #3: All comments have been addressed

2. Is the manuscript technically sound, and do the data support the conclusions?

Reviewer #2: Yes

Reviewer #3: Yes

3. Has the statistical analysis been performed appropriately and rigorously? 

Reviewer #2: Yes

Reviewer #3: Yes

4. Have the authors made all data underlying the findings in their manuscript fully available?

Reviewer #2: Yes

Reviewer #3: No

5. Is the manuscript presented in an intelligible fashion and written in standard English?

Reviewer #2: Yes

Reviewer #3: Yes

6. Review Comments to the Author

Reviewer #2: (No Response)

Reviewer #3: please address the following issues that I would like you to effect before resubmitting:

1. Your current structure is messy and so please re-structure paper as follows: 1. Introduction, 2. Background, 3. Theoretical literature review, 4. Empirical literature review and hypotheses development, 5. Research design, 6. Empirical results and discussion, and 7. Summary and conclusion. Please you must re-structure your paper according this comment 1, which will involve substantial revisions to your work. To reduce wasting the time of reviewers, yours and mine, if this suggestion is not followed, then, the paper may be rejected without any further review, and so please kindly take this comment seriously.

2. Introduction: Please clarify your research questions, objectives, background motivation, theoretical and empirical motivation and the lines of contributions to the literature. You can do this by sharply articulating your research questions/objectives, identify the potential theoretical, background and theoretical motivation or gaps, and explain how your study contributes to the literature. You can do this by highlighting the weaknesses of prior studies as well. Currently, your introduction is very dry. Additionally, you need state clearly the contributions of the paper. For example, “Consequently, the current paper seeks to make the following contributions to the existing literature. First,…, Second,…., Third, …, Fourth,… and so on”. The introduction should be about 5 pages long.

3. Background – you need to explain why this is the appropriate context to conduct this study by exploiting regulatory, reform and policy issues and developments within the research context or setting. This should be about two to three pages long.

4. Theoretical framework - Please an overarching theoretical framework that will explain the underlying predictions and hypotheses of interest. In doing so, please explicitly outline how they help link the dependent and independent variables together by drawing on both seminal (old) and recently (newly) published studies. This should be about two to three pages long.

5. Literature review and hypotheses Development – please enhance your hypotheses by: (i) drawing on the theory; (ii) empirical literature; (iii) research setting/contextual insights; and (iv) then setting up your hypotheses. You will do this for each hypothesis. Currently, you have not developed your hypotheses in this way. You will need to so by drawing on both seminal (old) and recently (newly) published studies.

6. Research design – Please identify, classify and explain your variables – dependent, independent and control variables, as well as any others, such as moderating or mediating variables. Please also explain your sample selection clearly (insert a table tabulating the steps - how many was missing, many had data, how many selected and why) and also clarify in a normative way how the variables are operationalised. Similarly, explain your sample in a tabular form, outlining step by step the total population to the selection of the final sample. Label all your equations, figures and tables in a consecutive manner. Make the tables self-contained by clearly identifying dependent, independent and control variables in the tables.

7. Empirical findings – please link your findings more strongly to the: (i) theory, (ii) empirics, (iii) context; and (iv) highlight their economic, academic/research and policy implications. Closely link up and cite the papers that you have discussed in the background, theory and empirical literature review & and hypotheses development section to the findings you are presenting here.

8. Conclusion – Please outline a summary of findings, contributions, implications, limitations and avenues for future research. Especially, expand the discussions relating to implications, limitations and avenues for future research.

9. Robustness or additional analyses – please demonstrate how your findings are to alternative measures (e.g., different ways of measuring the key dependent and independent variables), estimations (e.g., lagged structure, and instrumental variables estimation, amongst others) and general endogeneities. This is completely missing.

I hope you will positively embrace these constructive suggestions as a way of taking this research forward, and I look forward to receiving a revised version of your paper.

Best regards

7. PLOS authors have the option to publish the peer review history of their article (what does this mean?). If published, this will include your full peer review and any attached files.

Reviewer #2: No

Reviewer #3: No

---

## [Author Response · Author response to Decision Letter 1]

10 Jun 2023

See the enclosed Letter

(I also copy the content here)

First of all, let us thank the Reviewers and the Editor, once again, for their very careful reading of this article and valuable comments. We also thank them for the encouraging evaluations.

We received no comments from Referee 1 and 2 (Referee 2 was satisfied by the changes made in the revised version following his/her previous comments). Referee 3 made no comments regarding the “substance” of the research, while he/she recommended a deep revision of the exposition and presentation, with detailed suggestions regarding the structure of the paper. (However, some suggestions of Referee 3 are in conflict with the opinion of the Referee 1 on the first version of the paper --for instance, concerning the content of the Introduction). In any case, we paid serious attention on the comments of the Referee 3, and the structure of the paper has been changed accordingly. 

Specifically, the paper has been re-structured as follows. (1) Introduction, including: (i) a sub-section dealing with the back-ground of the case study; (ii) a sub-section with a review of the relevant literature; (iii) a sub-section explaining the specific contribution of the present study. (2) Theoretical framework and the research design. (3) Data. (4) Empirical findings, articulated in (i) The empirical model for the price dynamics; (ii) Daily seasonality in price level and volatility; (iii) Relations with financial markets; (iv) Relations with the exchange rate. (5) Additional analysis (with the computation of a price index from a hedonic price approach, which is also interpreted as a+ robustness check). (6) Conclusions.

This structure is substantially in line with the suggestion of Referee 3, and also consistent with the observations of Referee 1 in the first stage of the reviewing process.

We are confident that the new structure permits to highlight the contribution of the present analysis in a clear and effective way.

Let us notice that all the data used in the analysis are freely downloadable from the web; in any case, we also up-load the file with the (final) dataset used in the analysis. We can publish this file in open sites (and the journal has been already provided with the file).

---

## [Editor Report · Decision Letter 2]

16 Jun 2023

On the price dynamics of non-fungible tokens: The ‘Bored Apes’ case

PONE-D-22-31290R2

Dear Dr. Cellini,

We’re pleased to inform you that your manuscript has been judged scientifically suitable for publication and will be formally accepted for publication once it meets all outstanding technical requirements.

Kind regards,

Vasileios Kallinterakis

Academic Editor

PLOS ONE
---

## [Editor Report · Acceptance letter]

28 Jun 2023

PONE-D-22-31290R2 

On the price dynamics of non-fungible tokens:
The ‘Bored Apes’ case 

Dear Dr. Cellini:

I'm pleased to inform you that your manuscript has been deemed suitable for publication in PLOS ONE. Congratulations! Your manuscript is now with our production department. 

Kind regards, 

on behalf of

Dr. Vasileios Kallinterakis 

Academic Editor

PLOS ONE